# Microbial Diversity of Traditional Livno Cheese from Bosnia and Herzegovina

Tarik Dizdarević [1], Svijetlana Sakić-Dizdarević [1], Davide Porcellato [2], Zlatan Sarić [1], Mersiha Alkić-Subašić [1], Roger K. Abrahamsen [2] and Judith A. Narvhus [2,*]

[1] Faculty of Agriculture and Food Sciences, University of Sarajevo, Zmaja od Bosne 8, 71000 Sarajevo, Bosnia and Herzegovina; t.dizdarevic@ppf.unsa.ba (T.D.); s.sakic-dizdarevic@ppf.unsa.ba (S.S.-D.); z.saric@ppf.unsa.ba (Z.S.); m.alkic-subasic@ppf.unsa.ba (M.A.-S.)

[2] Faculty of Chemistry, Biotechnology and Food Science (KBM), Norwegian University of Life Sciences (NMBU), P.O. Box 5003, N-1432 Ås, Norway; davide.porcellato@nmbu.no (D.P.); roger.abrahamsen@nmbu.no (R.K.A.)

[*] Correspondence: judith.narvhus@nmbu.no; Tel.: +47-64-96-5000

**Abstract:** Traditional dairy products, especially cheeses, represent part of the cultural food heritage of many countries. In addition, these cheeses constitute microbiological "reservoirs", of which many have been lost due to the introduction of the pasteurization of milk in the dairy industry. Increased awareness of the importance of microorganisms that make up the biodiversity of traditional cheeses, as well as the development of molecular methods in recent decades, have enabled efforts to identify and preserve them. Traditional Livno cheese is a full-fat hard cheese, considered one of the most famous traditional cheeses of Bosnia and Herzegovina and is seasonally produced from a mixture of raw sheep's milk supplemented with cow's milk. Often, Livno cheese has variable quality, due to microbial contamination and poor milk quality. In this study, traditional Livno cheese was studied during the ripening of cheeses produced by different producers during two seasons. Culture-dependent analyses were made during ripening using microbiological plating on suitable media. Likewise, culture-independent methods Denaturing Gradient Gel Electrophoresis (DGGE) and Automated Ribosomal Intergenic Spacer Analysis (ARISA) were used to elucidate the cheese microbiota. Results of analysis showed *Lactococcus* spp., *Streptococcus* spp., *Lactobacillus* spp., *Pediococcus* spp. and *Leuconostoc* spp. to be dominant species in traditional Livno cheese. However, when comparing the use of culture-dependent and culture-independent methods in the evaluation of Livno cheese microbiota, *Enterococcus* was not detected by culture-independent DGGE methods. The microbial population of both the milk and the environment determines the fermentation processes during cheese production and ripening, and thereby defines the quality of this cheese. The numbers of bacteria in the cheese were shown to be dependent on the manufacturer, the degree of ripening and the production season.

**Keywords:** raw-milk Livno cheese; microbial diversity; spontaneous fermentation; ripening

## 1. Introduction

Traditional Livno cheese belongs to the group of hard cheeses produced from predominantly raw sheep's milk added varying amounts of cow's milk, without the addition of a starter culture. This cheese has been produced in the Livno area for more than 130 years. Production of this cheese is seasonal (from May to October), when the sheep are kept in the mountains surrounding the town of Livno, in the southwestern part of Bosnia and Herzegovina.

The quality of traditional cheeses depends on many factors including milk quality, cheese processing, diversity of herd management, as well as cheese-making and cheese-ripening practices. In many cases, such small-scale production has generated the diversity of characteristics in ripened cheeses that still exist in traditional cheeses [1]. One probable

explanation for this phenomenon could be that traditional cheeses are manufactured with the help of spontaneous fermentation and are therefore the result of the metabolic activity of a complex microbial ecosystem whose diversity and level of organization in such products is still largely unknown [2].

Raw milk cheese technology, without the addition of starter cultures, considerably depends on the microbial composition of the milk and the environment and often provides cheese of variable quality. These microorganisms are responsible for acidification of the milk, as well as for many biochemical reactions during production and ripening, thus developing the sensory properties of cheese. Simultaneously, the process of cheese ripening results in numerous interactions within the microbial cheese community, and the taxonomic identification of the microorganisms is of great importance for understanding these processes [3,4].

Traditional raw milk cheeses are characterized by a complex microbiota. The microbial flora of artisanal cheeses is a pool of autochthonous lactic acid bacteria (LAB). It may be possible to select microbes that have a good potential for use as a starter culture during milk acidification or as a secondary microbiota during cheese ripening in industrial level with the aim of obtaining cheeses with sensory characteristics similar to those of artisanal cheeses [5,6]. LAB are Gram-positive, catalase-negative, non-spore-forming cocci or rods. These microorganisms play an important role in food fermentations and are characterized by the production of lactic acid as the main metabolic product. In addition, they may produce secondary metabolites, such as bacteriocins, hydrogen peroxide, and diacetyl, which can prevent the growth of unwanted bacteria as well as food spoilage pathogens. During the production and ripening of cheese, a complex interaction takes place between starter LAB (usually deliberately added for curd acidification) and non-starter LAB (NSLAB, adventitious milk contaminants from farm and dairy environments) from milking to ripening [7–9].

The results of research on 35 European cheeses showed a complex microbial biodiversity in the cheese regardless of whether culture-dependent or -independent methods were used; *Lactococcus* spp. accounted for 38%, *Enterococcus* spp. 17%, *S. thermophilus* 14%, mesophilic *Lactobacillus* 12%, *Leuconostoc* spp. 10% and thermophilic *Lactobacillus* 9% of the total microbial population [10].

The presence of certain microbial species during cheese production and ripening is largely determined by the fermentation processes that define the ultimate quality of the cheese. It is believed that traditional cheeses, made from raw milk, have a more intense taste and flavor when compared to cheeses industrially produced from pasteurized milk [11,12].

Historically, culture-dependent methods have been used for the study of microbial communities associated with fermented foods. However, these methods have established limitations for difficult-to-culture microorganisms [13]. The reasons for this are numerous and include pH, substrate composition, oxygen availability, or that cells are viable but non-culturable. Such cells are metabolically active but lack the ability to grow on laboratory growth media [14,15]. Some cultivation techniques may underestimate the microbial diversity present or even fail to detect some major groups of microbes. To overcome these cultivation problems, several culture-independent molecular methods based on the amplification of nucleic acids by polymerase chain reaction (PCR) have been developed [16].

In the last few decades, cultivation-independent molecular methods based on the direct isolation of microbial DNA from the sample to be studied have been developed. Amplification of the target gene with PCR, followed by post-PCR analysis of genetic material using various techniques have been increasingly used. Some of these methods include RAPD (randomly amplification of polymorphic DNA), ARISA (automated ribosomal intergenic spacer analysis), T/DGGE (temperature/denaturing gradient gel electrophoresis) [17–19].

The PCR-DGGE method is a commonly used molecular method for testing microbial biodiversity [2]. The ARISA method has only recently been applied to the study of the microbial biodiversity of cheese [20–23], as well as for testing the microbial biodiversity of other media such as dough [24], wine [25], biofilm [26], etc.

The microbial biodiversity in cheese is, to an increasing degree, now analyzed by means of next-generation sequencing (NGS) analysis that provides a comprehensive description of the DNA content of microorganisms in a tested sample, generating up to $10^9$ sequences [27,28]. However, the disadvantage of cultivation-independent methods is the inability to isolate individual microbial cultures for subsequent application as starter cultures or secondary ripening organisms.

The aim of this research was to examine the microbial biodiversity of the traditional Livno raw milk cheese made by three traditional small cheese producers who are also livestock farmers. Cheese samples were taken during two consecutive years (seasons) when sheep and cows are grazing in the mountains. Variations between production seasons, producers and ripening periods of traditional Livno cheese were analyzed. In order to determine the microbial composition of traditional Livno cheese, both culture-independent and culture-dependent methods were applied.

## 2. Materials and Methods

### 2.1. Traditional Livno Cheese Production and Sampling

Traditional Livno cheese belongs to the group of hard-type cheeses, has cylindrical form, straw-yellow color and weighs between 2–3 kg. In this study, Livno cheese was produced from a mixture of raw sheep's and cow's milk (evening and morning milkings). Milk was heated to 31–33 °C and natural calf rennet powder was added for coagulation (curdling time was around 50 to 75 min). After coagulation, curd was cut into about 8 cm cubes and then granulated to the size of a kernel of wheat. Thereafter, curd grains were heated to 45–48 °C for 20 to 40 min with continuous stirring of the curd. The cheese curd was placed into the molds lined with jute cheesecloth, pressed for 12 to 24 h with constant turning of the cheeses. After pressing, cheese was salted in a brine (22–25% NaCl) for 48 h. When brining was finished, cheeses were left to ripen at a temperature of 12–16 °C and humidity of 70–85% for up to 90 days. The usual ripening lasts between six and twelve weeks.

A total of 30 samples of traditional Livno cheese and curd were taken from three different producers (A, B, C) from the major cheese-producing area—the Livno region—during two consecutive years (two production seasons—1st; A, B, C; 2nd; a, b, c).

During each production season, cheese samples were taken in the same way and transported at 4 °C to the Faculty of Agriculture and Food Sciences University of Sarajevo (FAFS).

At each producer, one fresh curd (Cu) sample was taken from the cheese vat after milk coagulation and stored aseptically in a sterile jar. Similarly, one block of cheese was sampled after pressing and salting, before ripening (1st day) at each producer. In addition, three cheeses from the same batches were left to ripen for up to 90 days under controlled conditions of 16 °C and 85% humidity.

Fresh curd samples (Cu) after coagulation as well as cheese after pressing and salting, but before ripening (1st day), were analyzed immediately upon delivery to the FAFS. During the ripening period, one cheese block from each batch was analyzed after 30, 60 and 90 days of ripening.

All 30 cheese and curd samples were divided in two parts; one part of the sample was immediately plated on different agar media and the other part was stored at −80 °C and transported in dry ice to the Norwegian University of Life Sciences—NMBU (Ås, Norway). Here, the cheese microbiota was analyzed using culture-dependent and -independent methods.

### 2.2. Culture-Dependent Microbial Analysis

Culture-dependent methods were used for the determination and isolation of microbial population in Livno cheese curd, fresh salted cheese (1 day) and cheeses during ripening (30, 60 and 90 day). Different nutrient media were used. A 20 g sample of curd or cheese was mixed with 180 mL sterile aqueous solution of 3% Na-citrate, at 45 °C for 3 min using

the BagMixer 400 W (Interscience, St. Nom, France). From this basic dilution, decimal dilutions were prepared using sterile $\frac{1}{4}$ strength Ringer's solution (Merck, Darmstadt, Germany). Milk plate count agar (Liofilchem, Roseto degli Abruzzi, Italy) was used for the enumeration of the total count of aerobic bacteria in samples, after incubation for 72 h at 30 °C. Presumptive mesophilic and thermophilic lactobacilli were determined and isolated on MRS agar (Merck) adjusted to pH 5.4 and incubated anaerobically (10% $CO_2$) at both 25 °C and 45 °C for 72 h and 48 h. The determination and isolation of presumptive mesophilic lactococci and thermophilic streptococci was made by aerobic incubation at 25 °C and 45 °C for 72 h using M17 agar (Merck). *Leuconostoc* spp. were determined using MRS agar supplemented with 20 mg/L$^{-1}$ of Vancomycin incubated for 72 h at 30 °C. *Enterococcus* spp. was determined on Slanetz and Bartley agar (Merck) after incubation for 48–72 h at 37 °C. A total of 30 cheese curd and cheese samples were tested. For each sampling time, 6 samples were analyzed.

### 2.3. Culture-Independent Microbial Analysis

Denaturing Gradient Gel Electrophoresis (DGGE) and Automated Ribosomal Intergenic Spacer Analysis (ARISA) methods were applied.

### 2.3.1. Extraction and Purification of Total DNA from Cheese

Total genomic DNA was extracted from curd, fresh salted cheese and traditional Livno cheese after 30, 60 and 90 days of ripening. From the cell pellet, the total DNA was extracted and purified [29], using the commercial GenElute Bacterial Genomic DNA kit (Sigma-Aldrich, St. Louis, MI, USA). After extraction, the DNA was kept frozen at −20 °C until analysis.

### 2.3.2. PCR-DGGE Analysis and Band Identification by Sequencing

PCR reactions were performed in a final volume of 20 μL using 1× LightCycler® 480 HRM MasterMix (Roche, Mannheim, Germany), 2 mmol L$^{-1}$ of MgCl$_2$, 0.4 μmol L$^{-1}$ of each primer and 1 μL of DNA extracted from cheese samples. Two combinations of primers were used per sample. The primer pair LAC1 (5′-AGCAGTAGGGAATCTTCCA-3′) and LAC2 (5′-ATTTCACCGCTACACATG-3′) were used to identify microorganisms from the genera *Lactobacillus* (L.), *Leuconostoc* (Leu.) and *Pediococcus* (P.) whereas for *Lactococcus* (Lc.), *Enterococcus* (E.) and *Streptococcus* (S.) spp., the primer pair LAC3 (5′-AGCAGTAGGGAATCTTCGG-3′) and LAC2 was used [30,31]. These primers were synthesized by Invitrogen Ltd. (Paisley, UK) and amplify the V3 region of the 16 S rRNA gene. Control strains from the laboratory collection (IKBM, NMBU) *Lactococcus lactis* subsp. *lactis*, *E. faecalis*, *Lacticaseibacillus paracasei* and *Lactiplantibacillus plantarum* were used during the PCR reaction and DGGE analysis. Isolated and purified DNA from curd and cheese was amplified using a quantitative qPCR LightCycler 480 Real-Time (Roche, Grenzach, Germany), using initial denaturing program 95 °C for 5 min, 30 denaturation cycles at 95 °C for 30 sec, annealing at 61 °C for 30 sec and elongation at 72 °C for 1 min. PCR products were separated using the DGGE INGENYphorU system (Ingeny International BV, Goes, The Netherlands). A total of 20 μL of PCR product with 4 μL color 6 × Gel Loading Dye (New England Biolabs Inc., Ipswich, MA, USA) were loaded into wells in a denaturing gel containing a gradient of 30–55% of urea-formamide as previously described [29].

The DGGE gel electrophoretic parameters were 75 V, 50 mA, 4 W and 60 °C for 16 h. After separation, the gel was dyed for 30 min in a solution of 1 × TAE buffer with 0.5 mg/L$^{-1}$ GelRed™ 3× (Biotium, Fremont, CA, USA) and visualize on the Gel Doc ™ transilluminator (Bio-Rad, Hercules, CA, USA). Bands of interest were excised with a sterile knife and placed in sterile Eppendorf tubes containing 50 μL of 0.1 × TE buffer. After incubation at 37 °C for 4 h to extract the DNA from the gel, 2 μL of the extract PCR product was re-amplified by the same PCR conditions used for the DGGE. The resulting PCR product was re-purified using a QIAquick PCR Purification Kit (Qiagen, Venlo, The Netherlands), and sequenced (GATC biotech AG, Konstanz, Germany).

The obtained sequence length of about 300 base pairs was used to determine microbial species with the help of BLAST program at NCBI GenBank (http://www.ncbi.nlm.nih.gov, accessed on 28 October 2023).

### 2.3.3. ARISA

Species of LAB possess multiple ribosomal operons that can vary in the length of the intergenic transcribed spacers (ITS) due to the presence of the tRNA between the 16S and the 23S rRNA gene. This phenomenon was used to evaluate the total amount of operational taxonomic units (OTUs) and their relative abundance in samples [21]. To determine the microbial diversity by ARISA method, a 16S–23S intergenic spacing database of common LAB found in cheese was used.

The ARISA was carried out according to the method in [32], using the set of primers [33]. One microliter of PCR products of total DNA was denatured with 10 µL highly deionized formamide (Applied BioSystems, Carlsbad, CA, USA) and mixed with 0.025 µL of standard GeneScan Liz 1200 (Applied BioSystems, Carlsbad, CA, USA). The samples were analyzed by capillary electrophoresis using the ABI 3700 Genetic Analyzer at the Institute of Molecular Medicine Finland (FIMM, Helsinki, Finland). The length and area of the obtained peaks were analyzed with Peak scanner software v.1.0 (Applied BioSystems, Carlsbad, CA, USA). Control strains used for the ARISA method in this case were the isolates obtained from the traditional Livno cheese after species were identified by using 16S rRNA sequencing (described below). Positive controls were *P. pentocaceus*, *P. acidilactici*, *Lc. garvieae*, *L. paracasei*, *L. plantarum*, *Lc. lactis* subsp. *lactis*, *E. faecium*, *Streptococcus macedonicus* and *E. faecalis*.

Data analysis for the cheese microbiota using ARISA method was performed against a database of ITS sequence length previously created. Analysis of similarity was used to identify differences between treatments and batches using the statistical software R (www.r-projekt.org, accessed on 28 October 2023).

### 2.4. Statistical Analysis

All the analyses were carried out in duplicate, with the production season being treated as a repeat. Significant differences between the producers and during the ripening period were evaluated by one-way analysis of variances (ANOVA), followed by the Student's *t*-test. The differences were considered significant at $p < 0.05$ using the software package SPSS version 20.0 (SPSS Inc., Chicago, IL, USA).

## 3. Results

### 3.1. Enumeration of Bacteria Using Culture-Dependent Methods

Seven different nutrient media were used to investigate the microbial diversity of traditional Livno cheese by cultivation-dependent methods: total count of aerobic bacteria, presumptive mesophilic and thermophilic streptococci, presumptive and thermophilic mesophilic lactobacilli, *Leuconostoc* spp. and *Enterococcus* spp. The mean values (log10 CFU·g$^{-1}$) of the microbial composition of traditional Livno cheese, produced in two seasons by three producers (A, B, C—first season; a, b, c—second season), during 90 d of ripening after growth on various selective substrates, are presented in Table 1.

The LAB counts showed no significant differences between three producers ($p > 0.05$) Production season had a significant effect ($p < 0.05$) on the numbers of all examined microbial groups, except for *Enterococcus* spp.

Ripening period had a significant effect on all observed microbial groups ($p < 0.05$), except for *Enterococcus* spp. ($p > 0.05$). The number of presumptive mesophilic lactococci ($p < 0.05$) and presumptive thermophilic streptococci ($p < 0.03$) were lowest in curd samples, and highest in cheese samples after 1 day. After the 1st day, most of the tested microbial groups decreased during the ripening period observed up to 90 days.

**Table 1.** Microbial count (log10 CFU·g$^{-1}$) in curd (Cu) and cheese after 1, 30, 60 and 90 days of ripening of traditional Livno cheese, from three producers and two seasons.

| | Days | Total Count of Aerobic Bacteria | | Presumptive Mesophilic Lactococci | | Presumptive Thermophilic Streptococci | | Presumptive Mesophilic Lactobacilli | | Presumptive Thermophilic Lactobacilli | | *Leuconostoc* spp. | | *Enterococcus* spp. | |
|---|---|---|---|---|---|---|---|---|---|---|---|---|---|---|---|
| | | 1.season | 2.season | 1.season | 2.season | 1.season | 2.season | 1.season | 2.season | 1.season | 2.season | 1.season | 2.season | 1.season | 2.season |
| | Cu | 6.1 | 6.7 | 6.0 | 6.9 | 5.3 | 6.0 | 6.5 | 6.0 | 3.9 | 4.5 | 4.1 | 4.3 | 4.9 | 5.0 |
| | 1 | 8.9 | 10.2 | 8.7 | 9.7 | 8.3 | 8.7 | 7.0 | 7.3 | 6.9 | 8.4 | 7.1 | 8.3 | 6.4 | 8.0 |
| Producer A | 30 | 7.3 | 9.4 | 7.4 | 8.6 | 7.4 | 8.9 | 6.4 | 7.6 | 5.7 | 7.2 | 6.1 | 7.4 | 4.9 | 6.4 |
| | 60 | 6.8 | 8.8 | 6.4 | 8.6 | 5.9 | 8.8 | 6.2 | 7.3 | 6.1 | 5.6 | 6.5 | 7.5 | 6.3 | 7.1 |
| | 90 | 7.8 | 8.7 | 6.9 | 8.9 | 6.9 | 9.1 | 6.7 | 7.8 | 6.7 | 4.9 | 7.3 | 7.7 | 6.9 | 8.1 |
| | Cu | 6.2 | 5.9 | 7.1 | 6.6 | 6.4 | 6.6 | 6.9 | 6.8 | 4.5 | 6.3 | 4.8 | 4.0 | 6.4 | 6.8 |
| | 1 | 7.8 | 10.4 | 7.7 | 9.6 | 7.4 | 9.4 | 7.6 | 7.6 | 5.8 | 7.6 | 8.2 | 7.9 | 6.9 | 8.1 |
| Producer B | 30 | 7.9 | 9.1 | 8.6 | 8.4 | 5.7 | 8.9 | 7.8 | 8.0 | 5.8 | 6.6 | 7.8 | 7.4 | 6.8 | 7.9 |
| | 60 | 8.0 | 8.3 | 7.7 | 8.2 | 6.6 | 8.4 | 7.4 | 7.2 | 5.2 | 7.2 | 7.1 | 7.2 | 5.7 | 7.5 |
| | 90 | 7.5 | 8.2 | 7.2 | 7.5 | 7.2 | 8.4 | 7.2 | 7.2 | 5.4 | 6.8 | 7.3 | 7.3 | 5.3 | 7.3 |
| | Cu | 6.3 | 7.0 | 6.7 | 6.9 | 6.9 | 5.9 | 5.9 | 5.5 | 4.1 | 4.4 | 5.8 | 5.0 | 5.9 | 7.1 |
| | 1 | 9.8 | 8.7 | 8.6 | 9.0 | 9.0 | 9.1 | 7.1 | 7.5 | 6.9 | 6.9 | 7.4 | 7.6 | 6.4 | 8.4 |
| Producer C | 30 | 7.7 | 9.2 | 7.8 | 8.3 | 6.4 | 8.8 | 7.4 | 7.6 | 7.2 | 6.3 | 6.7 | 7.3 | 7.0 | 6.9 |
| | 60 | 9.4 | 8.8 | 6.5 | 8.1 | 6.9 | 7.9 | 6.7 | 7.3 | 6.7 | 6.4 | 6.0 | 7.4 | 6.7 | 7.4 |
| | 90 | 8.0 | 8.7 | 7.9 | 7.2 | 7.9 | 8.3 | 7.4 | 7.3 | 7.8 | 6.7 | 7.9 | 6.9 | 7.9 | 7.3 |

### 3.2. Microbial Diversity of Traditional Livno Cheese by PCR-DGGE

DNA for DGGE analysis was extracted from 30 samples of cheese curd and traditional Livno cheese during ripening. Samples from two production seasons (first and second) and from each of three producers (A/a, B/b, C/c) were examined.

By excising the bands on the PCR-DGGE gel, followed by reamplification and sequencing, 30 samples of traditional Livno cheese and curd showed dominant microbial groups belonging to the following species: *Lc. lactis* (band 11), *S. thermophilus* (band 10), *S. macedonicus/gallolyticus/parauberis* (band 14, 13 and 12), *L. plantarum* (band 3) and *L. paracasei* (band 1). *L. plantarum* was detected in Livno cheese samples from all three producers in the second season, while in the first season its presence was seen only at producer A.

DGGE analysis of curd and traditional Livno cheeses at different ripening stages identified a total of 15 bands that corresponded to different microbial species (Figure 1). The *Lactobacillus*-specific gel showed 10 bands, whereas on the gel for lactococci, 5 further bands were found. The number of bands in each sample ranged from 3 to 9, depending on producer, season and ripening stage. The results of all microbial species that were found in traditional Livno cheese are summarized in Table 2.

*L. helveticus* (band 2) was identified in samples A and C, with slightly more intense bands in the second season. *Lactiplantibacillus plantarum* (band 3) was visible in the cheese curd from producer A, while in the second season it was determined only in the cheese sample after 30 days of ripening. This band was not shown in cheese from producer B in the first season but was found in all stages of ripening in the second season. Similar results were obtained from the samples from producer C, but in the first season it was detected only in the curd sample and was present in all cheese samples during the ripening season.

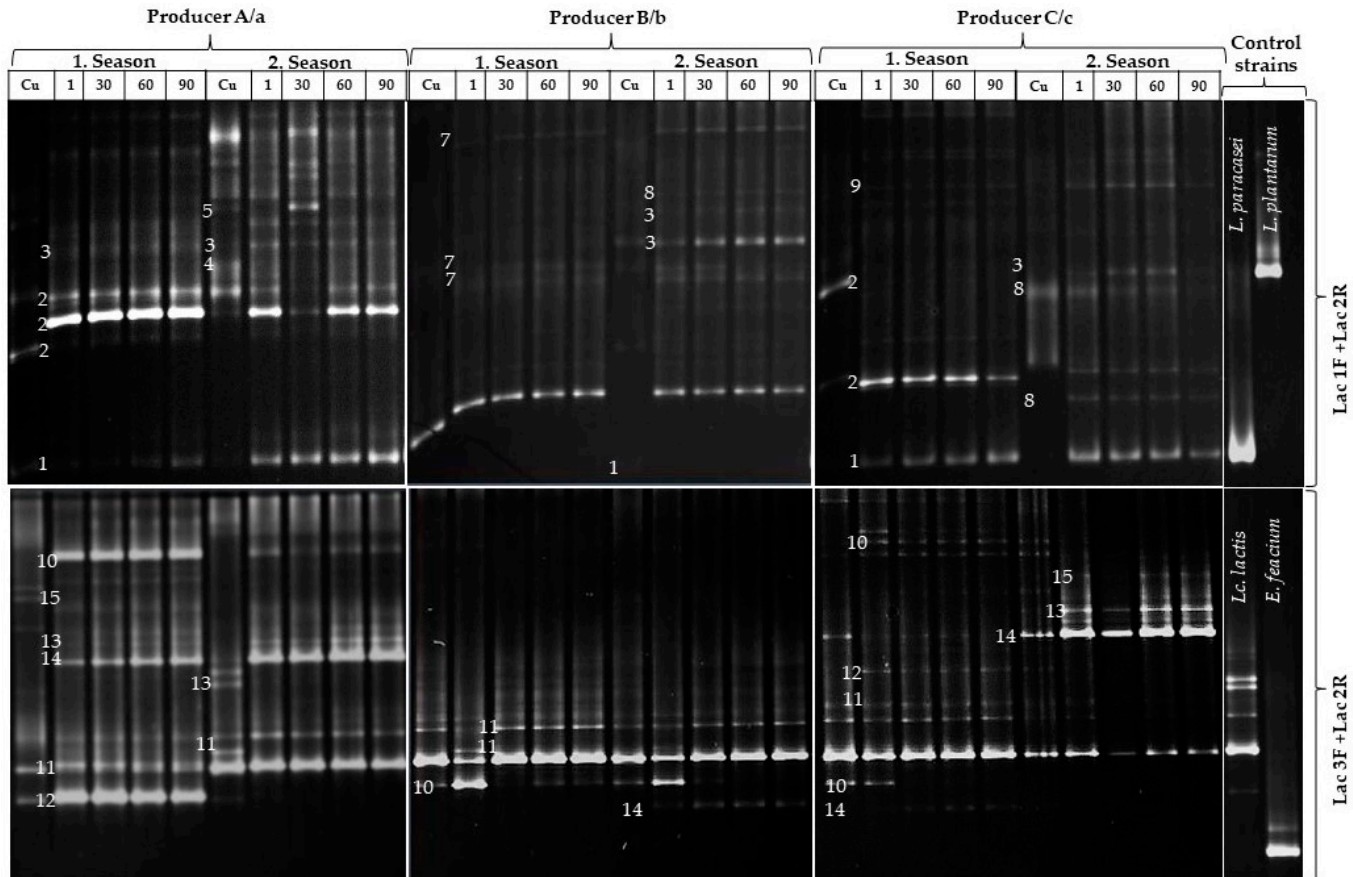

**Figure 1.** DGGE profile showing the dynamics of bacteria of traditional Livno cheese in fresh curd (Cu), at 1st (1), 30th (30), 60th (60) and 90th (90) day of ripening at three producers (A/a, B/b, C/c), in two seasons (1st and 2nd).

**Table 2.** Identification of dominant bacteria in traditional Livno cheese during manufacture and ripening for 90 days based on rRNA gene sequences obtained by DGGE.

| Producer [1] | Band | Identification [2] | NCBI-BLAST Accession Number Access Date: 28 October 2023 | Identity (%) |
|---|---|---|---|---|
| A, B, C, a, b, c | 1 | *Lacticaseibacillus paracasei* | OR267407.1 | 99 |
| A, C, a, c | 2 | *Lactobacillus helveticus* | MT538439.1 | 98 |
| A, B, C, a, c | 3 | *Lactiplantibacillus plantarum* | EF597125.1 | 98 |
| A, a | 4 | *Leuconostoc citreum* | ON631287.1 | 99 |
| a | 5 | *Leuconostoc pseudomesenteroides* | OM943117.1 | 98 |
| B, b | 6 | *Lactobacillus delbrueckii* subsp. *bulgaricus* | MT516034.1 | 99 |
| B, b | 7 | *Pediococcus pentosaceus* | OR518626.1 | 97 |
| A, C, a, c | 8 | *Leuconostoc mesenteroides* subsp. *mesenteroides* | OK135479.1 | 98 |
| C | 9 | *Lactobacillus rhamnosus* | CP101845.1 | 98 |
| A, B, C, a, b, c | 10 | *Streptococcus thermophilus* | KX926522.1 | 99 |
| A, B, C, a, b, c | 11 | *Lactococcus lactis* subsp. *lactis* | JN851797.1 | 99 |
| A, C, a, c | 12 | *Streptococcus parauberis* | MT597919.1 | 97 |
| C, c | 13 | *Streptococcus gallolyticus* | OP714497.1 | 97 |
| A, C, a, c | 14 | *Streptococcus macedonicus* | MN305794.1 | 96 |
| A, c | 15 | *Lactococcus garvieae* | OR502221.1 | 98 |

[1] Producer A, B, C in 1st season; (a, b, c in 2nd season); [2] Band identficiation according to Figure 1 (DGGE gels).

*Leu. citreum* (band 4) was found only in curd from producer A, whereas in the second season in the curd *Leu. pseudomesenteroides* was detected (band 5).

*L. delbrueckii* subsp. *bulgaricus* (band 6) was present in all samples from producer B but not in cheese from producers A and C. *P. pentosaceus* (band 7) was found in cheese from producer B, while *Leu. mesenteroides* (band 8) was found in cheeses from the second season in samples from producers B and C. Band 9 (*L. rhamnosus*) was determined only in cheese made by producer C from first season. *S. thermophilus* (band 10) was found in cheese samples from all three producers and microorganism also produced two bands. Band 11 was a multiband made up of several bands on the DGGE gel and was designated as *L. lactis* subsp. *lactis.* This band was present in all samples. In Figure 1 it can be seen that the intensity of the band 11 in the DGGE profile was uniform in producer B cheese samples from both seasons and in all stages of ripening compared to cheese samples from producers A and C. The bands from samples from producer A were more intense in the second season than in samples from manufacturer C in this season. Samples from producer C gave, however, more intense bands in the first season. Band 13, determined as *S. gallolyticus*, was present in samples from producer A during both seasons, while samples from producer C only showed this band in the second season and not in any sample from producer B. In contrast, the multiline band determined as *S. macedonicus* (band 14) in samples from producers A and C consisted of two bands on the lower and middle part of the gel. However, in samples from producer B, *S. macedonicus* did not show a band in the middle of the gel, but in the lower part of the gel. This band was present in all cheeses from producer C, but in the second season they were more intense in the middle of the gel. Band 15 was determined as *Lc. garvieae* and the band appeared in cheese samples from producer A in the first season, and from producer C in the second season and only at the beginning of the ripening period.

*3.3. Microbial Diversity of Livno Cheese Using ARISA*

The DNA isolated from samples of curd and cheese used for ARISA analysis was the same as the DNA used for the PCR-DGGE.

ARISA analysis of curd and cheese samples showed a complex and dynamic microbial biodiversity, as did PCR-DGGE. Dominant LAB determined were *Lc. lactis*, *S. thermophilus*, *Lc. garvieae*, *E. faecium*, *L. plantarum*, followed with: *Latilactobacillus sakei*, *Lc. garvieae*,

*E. faecalis*, *Levilactobacillus brevis*, *L. paracasei*, *Leu. pseudomesenteroides* and *P. acidilactici*. Figure 2 shows the most frequently occurring LAB.

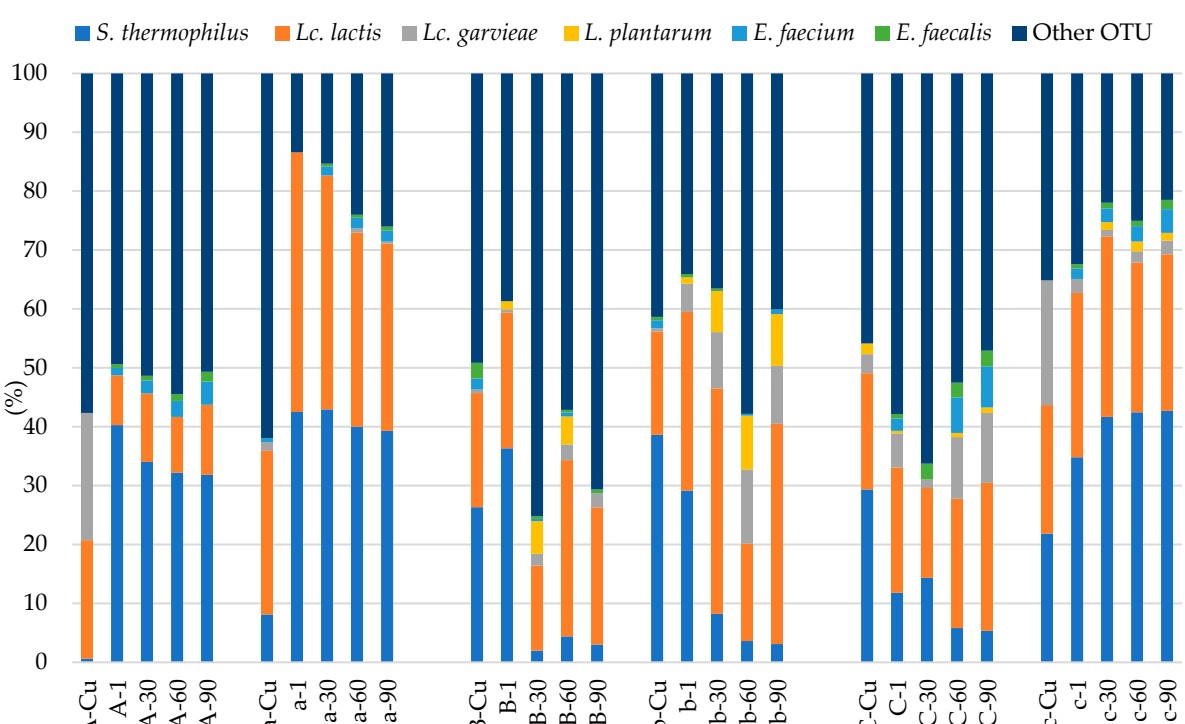

**Figure 2.** The most dominant operational taxonomic unit (OTU) determined by ARISA method. y axis shows the percentage of OTU that represents the microbial groups in the traditional Livno cheese from three producers and two seasons (A, B, C; a, b, c) and ripening time—days (Cu-curd, 1st, 30th, 60th and 90th day).

The results of the ARISA analysis, similar to the results of DGGE method, showed the dominance of *Lc. lactis* and *S. thermophilus* in all cheese samples. In addition to the dominant species, the slightest differences between species were determined depending on producer or season. However, the large proportion of "other OTU's ", occurring in the ARISA method, was disconcerting.

The microbial profiles of samples from producer A were similar for both seasons. In the curd samples from the first season a dominance of *Lc. garvieae* compared to the second season was found. After salting, the proportion of *Lactococcus* ssp. notably decreased and remained at this level during ripening. At the same time the proportion of *S. thermophilus* in cheese from producer A reached its maximum on day 1. *E. faecalis* and *E. faecium* in cheese samples from producer A appeared on the first day of ripening.

In cheeses from producer b (second season) and from producer C (first season) a slightly higher proportion of *Lc. garvieae* and *Lc. lactis* was determined, and a lower proportion of *S. thermophilus* after 30 days of ripening, compared to the samples from producer A. The proportion of *Lc. garvieae* in the cheese samples was notably reduced after 30 days of ripening at producer A (first season) and c (second season). At the same time, cheeses from producer B had a higher proportion of *Lc. garvieae* when compared to producer A. *S. thermophilus* showed a maximum proportion in samples of curd and cheese after salting (B-1). The proportion of *S. thermophilus* in cheeses was similar in both seasons. *E. faecium* in the cheeses from the second season showed a higher proportion in cheese samples after salting (a and c producer) than in samples from the first season. *L. plantarum* was determined at producers B and C in both seasons, while not at producer A.

The microbial composition from producer C in the first season also showed a dominance of *S. thermophilus*, *Lc. lactis* and *Lc. garvieae* in the curd samples. However, after

cheese salting (Day 1), the proportion of *Lc. lactis* was reduced and this downward trend continued until the end of ripening while the proportion of *Lc. garvieae* increased. The average level of *E. faecalis* during the study season recorded a slight increase from day 1 to the end of ripening at 90 days. For *E. faecium*, the proportion showed the same trend as *E. faecalis*. During the second season, the proportion of *Lc. lactis* increased from curd samples (c-90), while the proportion of *Lc. garvieae* was similar to the cheese from the first season. The proportion of *S. thermophilus* in samples from the second season was somewhat lower than in the first season and was not present in the curd but it was detected in cheese after 90-day ripening. *Leu. pseudomesenteroides* was found in both cheeses from producer C and in the first season from producer B, but not in cheese from producer A. *L. sakei* and *Levilactobacillus brevis* were detected sporadically in cheese and in small relative amounts. The presence of *P. acidilactici* was shown only in some cheese samples from producer C.

## 4. Discussion

### 4.1. Enumeration of Bacteria on Agar in Livno Cheese

The averages of microorganisms enumerated ($\log_{10}$ CFU·g$^{-1}$) in traditional Livno cheese of the three producers for the first and second season, respectively, were: total count of aerobic bacteria on MPCA agar (8.1 and 9.1), presumptive mesophilic lactococci on M17-25 (7.6 and 8.5), presumptive thermophilic streptococci on M17-45 (7.2 and 8.7), presumptive mesophilic lactobacilli on MRS-25 (7.1 and 7.5), presumptive thermophilic lactobacilli on MRS-45 (6.4 and 6.7), *Leuconostoc* spp. on MRS + V agar (7.1 and 7.5) and *Enterococcus* spp. on SB agar (6.4 and 7.5).

The highest numbers of aerobic bacteria (MPCA) were recorded in cheeses from producer b during the second season, while in cheeses from producers a and c, growth was slightly lower, and a decreasing trend was recorded during whole ripening period. The average value of the total count of aerobic bacteria on MPCA agar was almost one log unit higher compared to the average values obtained in numerous studies of Pecorino cheese [34].

The values of presumptive mesophilic lactococci in Livno cheese varied the most between cheese curd and cheese 1 day old by approximately 2.2 $\log_{10}$ CFU·g$^{-1}$. The results of presumptive mesophilic lactococci were approximately same to those in 12 months old Pecorino Romano, Fiore Sardo and Cantrestato Pugliese cheese [35]. The presumptive thermophilic streptococci on the M17-45 had higher values in the second season and number of presumptive thermophilic streptococci was close to the values for French semi-hard Salers cheese from raw milk [36].

The average value of presumptive mesophilic lactobacilli in Livno cheese was 7.3 $\log_{10}$ CFU·g$^{-1}$. Values for presumptive mesophilic lactobacilli were approximate to the values obtained for 12 samples of Italian Pecorino cheese made from raw milk [37].

Possible reasons for the relatively high values of presumptive mesophilic lactobacilli in the samples of traditional Livno cheese and curd could be the use of raw milk, the environment, dairy equipment, and the primitive conditions during traditional production. This is confirmed by research in which NSLAB were successfully isolated from dairy equipment, walls, floors and milk-handling surfaces that were used daily in cheese production [1,38]. Likewise, results of testing Cheddar cheese showed that the number of NSLAB in curds varied between $10^2$–$10^3$ and gradually increased to a value of $10^7$–$10^9$ CFU·g$^{-1}$ after several months of cheese ripening [39].

As an important part of cheese microbiota, NSLAB consists of mesophilic and facultatively heterofermentative lactobacillus, such as *L. paracasei*, *L. casei*, *L. plantarum* and *L. rhamnosus*, but also pediococci and obligate heterofermentative lactobacillus. During the ripening of cheese, NSLAB are known to become more dominant than the starter organisms in the cheese, NSLAB has therefore the opposite kinetics of bacterial growth in cheese [40,41].

The average number of presumptive thermophilic lactobacilli on MRS agar was slightly lower compared with presumptive mesophilic lactobacilli but with similar trends during

ripening. The cheeses on the 1st day of ripening had the highest average values. Similar results of presumptive thermophilic lactobacilli are reported for 20 Italian cheeses, where mean values varied from 3.90 $\log_{10}$ CFU·g$^{-1}$ for Tuma Persa cheese to 8.10 $\log_{10}$ CFU·g$^{-1}$ for Vaccino cheese [42].

All examined microbial groups, determined on different nutrient agar media, except presumptive thermophilic lactobacilli (MRS-45) had a significant value in the second season. Possible reasons for different microbiota in raw milk cheeses without the addition of starter cultures can be affected by a range of biotic (natural fermentation, presence of contaminating microbes, biofilm, and microbial metabolites) and abiotic factors (technological processes, salt, environmental conditions) [4].

*Leuconostoc* spp. did not notably deviate from the value of other cheeses from raw milk [43] and in the study of traditional Spanish Cabrales, *Leuconostoc* spp. was found in a cheese curd at 4.60 $\log_{10}$ CFU·g$^{-1}$ to 6.14 $\log_{10}$ CFU·g$^{-1}$ in cheese after the 90th day of ripening. *Leuconostoc* spp. has been found in different types of cheese from raw milk. The *Leuconostoc* spp., *Leu. mesenteroides* and *Leu. mesenteroides* subsp. *dextranicum* were most often isolated from sheep's cheese from raw milk [44].

A relatively high number of *Enterococcus* was found in all Livno cheese and curd samples, in both seasons. These autochthonous dairy enterococci are often associated with traditional cheeses of the Mediterranean area [45]. During both seasons, cheeses from manufacturer A had higher enterococcus values compared to cheeses from manufacturers B and C. Moreover, a statistically significant difference was found between the average number of *Enterococcus* spp. and production season. In the first season, this was 6.3 $\log_{10}$ CFU·g$^{-1}$, while for the second season it was $\log_{10}$ 7.3 $\log_{10}$ CFU·g$^{-1}$. The values obtained were in the range from 5.0 to 7.0 $\log_{10}$ CFU·g$^{-1}$ reported for mature cheeses from Mediterranean countries [46]. A possible explanation for the difference in the numbers of *Enterococcus* spp. between the production seasons can be in the different weather conditions during production seasons, methods of animal breeding, hygienic conditions of processing, etc.

This "controversial" microorganism was considered for a long time to be an indicator of fecal contamination. However, recent studies have undermined these assumptions, and numerous authors give examples that point to the importance of these microorganisms, for instance for the development of cheese flavor and the production of bacteriocins in the cheese [47–49].

In Fiore Sardo, cheese enterococci made up a significant proportion of the total microbial biodiversity of this Italian cheese. It was found that the total number of enterococci in the milk used in the production of this cheese was 4.4 ± 0.59 $\log_{10}$ CFU·mL$^{-1}$ and that the number gradually increased during 30 days of ripening (to 7.03 ± 1.07 $\log_{10}$ CFU·g$^{-1}$) when it also reached its maximum. Further ripening revealed a gradual decrease in the number of enterococci, to 2.05 ± 2.21 $\log_{10}$ CFU·g$^{-1}$ after 6 months [50].

The number of enterococci in mature Slovenian Tolminc cheese ranged from 5.17 to 7.57 $\log_{10}$ CFU·g$^{-1}$. Out of a total of 118 isolates of enterococci the following were identified by PCR: *E. faecium* (71.2%), *E. durans* (20, 3%) and *E. faecalis* (8.5%). On the contrary, research of the enterococci population in artisanal Manchego cheese was found to be 81.8% *E. faecalis* of all isolates, while *E. faecium*, *E. hirae* and *E. avium* were present in low proportions [51,52].

The reason for the higher number of enterococci in Livno cheeses could be the higher initial number of enterococci in the milk and curd, caused by external contamination. Often, the problem in the technology of traditional Livno cheese is milking process of sheep, which is carried out by hand in an open area.

### 4.2. DGGE

Although molecular methods for the determination of microbial diversity have been intensively used for more than two decades, the microbial diversity of cheeses, especially those made from raw milk, is still an important area of research.

Based on the results of the analysis of PCR-DGGE Livno cheese from three producers, during ripening for 90 days and two seasons, a complex microbial profile was shown, which is in agreement with the results from other cheeses from raw milk, especially cheeses produced without the addition of starter cultures [53–56].

The DGGE bands of cheese samples from producer A and C showed a similar microbial composition, and producer B differed from the two other producers. A slight difference in DGGE profiles could be seen for samples from cheeses from the two seasons. This difference consisted of the presence or absence of individual bands, but the dominant bands of microbial populations were present in samples from both seasons. The smallest changes in the DGGE profile were observed during the ripening of the cheeses, while the most significant differences were found between the curd samples and the cheese on day 1, similar to that determined with the cultivation-dependent method.

In addition, a single microbial species may show as one or more bands positioned on different parts of the gel, such as *Lc. lactis* (band 11) or *L. helveticus* (band 2), possibly due to slight strain differences. The reproducibility of the PCR-DGGE profile method provides an excellent tool for the comparative analysis of the structure of microbial communities [57]. The bands obtained from DGGE were less distinct in curd samples when compared to cheese samples throughout the ripening period. With regards to DGGE results, the dominant microorganisms were *Lactobacillus* spp., *Leuconostoc* spp. and *Pediococcus* spp. The number of bands from DGGE gels generated with the LAC1-LAC2 primer was lower compared to LAC3-LAC2. A possible reason for the bands LAC3-LAC2 primer dominance might be explained by reduced conditions that lead to autolysis of microorganisms, whilst different products are released and used by NSLAB [57].

Curd samples bands were less bold, and the intensity of the bands increased with ripening progress, being strongest on the 90th day. Also, the intensity of the bands between the two seasons (first and second) was different. A possible explanation for this phenomenon could be the fact that curd samples had low dry matter in comparation to older cheeses. An identical phenomenon was recorded with the autochthonous Sicilian Ragusano cheese, where dramatic changes in the microbial composition between milk, curd and cheese during ripening occurred, with slightly higher positioning of the bands in the curd samples. The same authors point to the greater complexity of determining the microbial diversity of traditional cheeses compared to industrial cheeses [58,59].

In traditional Livno cheese, with a scalding temperature of 48 °C, the domination of presumptive thermophilic species was determined, as well as a remarkable number of mesophilic microorganisms, such as *Lc. lactis*. This can be explained by the resistance of wild-type lactococci microorganisms to elevated temperatures. In addition to numerous other microbial species, *Lc. lactis* subsp. *lactis* was dominant in Istrian cheese from raw sheep's milk made without the addition of starter culture [60]. The dominance of this microorganism is not expected, since cheese scalding is carried out at a temperature up to 42 °C for a duration of 10 to 15 min [61]. Similar results from DGGE analysis were obtained by examining the traditional Croatian fresh sheep cheese named Karakačanski skakutanac, which, in addition to *E. faecalis* and *Leu. pseudomesenteroides*, were dominated by *Lc. lactis* subsp. *lactis* [62].

The analysis of DGGE bands of Livno cheese did not show a single amplicon belonging to *Enterococcus* spp. Similar results were obtained by analysis of two traditional Iranian cheeses where the DGGE method did not detect amplicons corresponding to *Enterococcus* spp., despite being the dominant isolate by culturing during ripening [63].

The lack of detection of *Enterococcus* spp. in Livno cheese by this method can be explained by the sensitivity of DGGE. The sensitivity of this method, as well as the successful identification of microbial communities, depends not only on the type or even strain of microorganisms, but also on the presence of other microorganisms that enter the tested community, and the type of food matrix that is examined [64].

The DGGE can be considered a useful method for obtaining an initial profile of microbial diversity and can help in understanding the dynamics and structure of the microbial community, but this method also shows some limitations [65].

There are various reasons for the limitation of DGGE method, including gene interspecies heterogeneity, template annealing, temperature annealing, single DGGE bands not always representing single bacterial species, intraspecific polymorphisms and differential gene amplification [66].

*S. thermophilus*, *E. durans*, *Leuconostoc* spp., *L. casei* and *Lactobacillus rhamnosus* were also found using PCR-DGGE in Italian Pecorino Siciliano cheese produced from raw sheep milk, without the addition of starter cultures [67].

The presence of *Lactobacillus* spp. with dominance of *L. plantarum* and *L. kefiranofaciens* was reported for Castelmagno PDO semi-hard cheese, but also *Lc. lactis*, *Lc. lactis* subsp. *cremoris*, *S. agalactiae* and *Macrococcus caseolyticus* were identified. These species were determined by sequencing DGGE bands from DNA extracted from the samples of milk, curd and cheese during ripening. Also, in these cheeses, there was no notable variation between the number and position of bands in the DGGE profiles of analyzed samples of milk, curd and cheese [68].

A similar phenomenon was recorded in artisanal Pecorino Crotonese cheese, where dramatic changes occurred in the microbial composition between milk, curd and cheese during ripening. The same authors also underlined the greater microbial diversity of traditional cheeses compared to industrial cheeses [55].

It was reported that, depending on the period of the ripening, the dynamics of each population in Mexican artisanal Cotija cheese varied, whereby microbial flora identified in cheese could be linked with usage of the raw milk and cheese-making environment [57].

Multiple bands with 16S rRNA in the DGGE method were also observed, in cases where microbial species may have multiple copies of 16S rRNA and this phenomenon was considered a disadvantage in the profiling of microbial communities using the PCR-DGGE method [69].

### 4.3. ARISA

The results of the microbial profiles of traditional Livno cheese using the ARISA method showed a greater similarity between the cheeses of producers A and C compared to B and in this way were similar to the the results obtained with the DGGE analysis.

In all samples of Livno cheese, the dominance of *S. thermophilus* in the cheese curd was established. After pressing and salting of cheeses on the 1st day, the dominance of the OTU determined as *Lc. lactis* was established. A significant presence of *Lc. garvieae* and *L. plantarum* was found in the cheeses of producer B.

Although the DGGE method showed the presence of a band determined as *L. plantarum* in cheese samples of all three producers, by ARISA method its presence was determined for producer B and reduced level for producer C, while for producer A this species was not detected.

*Lc. lactis* dominated in samples from all three producers. All samples from producer B and C showed a notable amount of *Lc. garvieae*, while in the samples from producer A was found only in the curd samples.

The dynamics of the presence of microbial species during ripening, determined on a base of average results of OTU-s in curd and cheese samples from all three producers, indicated a trend of continuous growth of *L. plantarum*, *E. faecium* and *E. faecalis*, while the growth of *Lc. garviae* showed a trend of reduction *L. plantarum* (0.96%) and *L. paracasei* (0.16%) had low OTU values in all cheese samples, while by DGGE method were identified as dominant during ripening period.

*S. thermophilus* and *Lc. lactis* had the highest OTU in cheeses after salting (1st day). After the 1st day, a decrease was observed for *S. thermophilus* for both seasons, while *Lc. lactis* had similar values during ripening period in 1st and decrease in second season.

In total microbial diversity of cheese samples *S. thermophilus* was represented in higher amount at samples of producer C compared to B, but notably less in A.

Results of determination of microbial diversity for traditional Livno cheese by ARISA method were complementary to results obtained by the DGGE method, and with culture-dependent method.

The ARISA was more sensitive and had the ability to analyze a large number of samples in a short time interval, a limiting factor of this method is the inability to identify organisms that have not been introduced into the ITS Database of Interspace Regions (ITS Internal Transcription Spacer) as well as a relatively small number of known ITS sequences compared to the 16S rRNA gene databases [29].

## 5. Conclusions

Studies on the biodiversity of traditional raw-milk Livno cheese by culture-dependent and -independent methods has shown a complex microbial composition. The results of culture-independent methods (DGGE and ARISA) have shown the dominance of microorganisms of the species *Lactococcus* spp., *Streptococcus* spp., *Lactobacillus* spp., *Pediococcus* spp. and *Leuconostoc* spp. These methods showed greater similarity between cheeses from producers A and C compared to producer B, although producer B had a slightly lower representation of *S. thermophilus* and slightly higher presence of *L. plantarum*.

Nevertheless, the variation in microbial diversity in Livno cheese between the producers and between production seasons in the late ripening period was not large. This could indicate that the similar ripening conditions used by the three producers led to a converging microbiota as ripening progressed. However, strain variation and the presence of undetected microorganisms may cause a variation in the specific properties of the cheeses yet to be elucidated.

A possible solution for the standardization of traditional Livno cheese could be the thermalization of milk followed by the application of a starter culture made exclusively from the indigenous microorganisms of traditional Livno cheeses. Further technological research of autochthonous microorganisms isolated from traditional Livno cheese is warranted to determine their potential for the establishment of starter cultures that could be used in the future work with the technology of this cheese.

**Author Contributions:** Conceptualization, T.D. and D.P.; Methodology, D.P., Z.S., R.K.A. and J.A.N.; Validation, M.A.-S.; Formal analysis, T.D. and S.S.-D.; Investigation, T.D., S.S.-D. and M.A.-S.; Writing—original draft, T.D.; Writing—review & editing, Z.S., R.K.A. and J.A.N.; Project administration, S.S.-D. and Z.S.; Funding acquisition, R.K.A. and J.A.N. All authors have read and agreed to the published version of the manuscript.

**Funding:** This research was funded by Norwegian Ministry of Foreign Affairs for funding the research within the HERD project grant number 332160.

**Institutional Review Board Statement:** Not applicable.

**Informed Consent Statement:** Not applicable.

**Data Availability Statement:** The datasets presented in this study can be found in online repositories. The 16S rRNA gene sequence of DGGE bands used to support the findings of our study were deposited in the GenBank repository http://www.ncbi.nlm.nih.gov, accessed on 28 October 2023.

**Acknowledgments:** We are particularly grateful to the Norwegian Ministry of Foreign Affairs for funding the research within the HERD project and to colleagues from the Norwegian University of Life Sciences (NMBU).

**Conflicts of Interest:** The authors declare no conflict of interest.

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
