# Peer review of "Microbial Diversity of Traditional Livno Cheese from Bosnia and Herzegovina"

_fermentation, doi:10.3390/fermentation9121006_

Round 1
Reviewer 1 Report
Comments and Suggestions for Authors
The authors have done a good piece of work. However, a major revision is needed before publication. The paper lacks a detailed description of the cheese production technology, considering that this is an important aspect that can influence the microflora of the product, the authors should include a description of the production technology in the text.
Lines 101-102: What are the differences between the 3 productions? Which seasons were taken into account?
Line 173: Explain LAB
Line 209: Which LAB are mentioned in the table? Lactobacilli? explain
Line 273: Use either lactic acid bacteria or LAB, standardise in text and tables
Line 522: What does biofilm transmission mean? Can only be transmitted by biofilm formation?
I advise the authors to revise and implement the conclusions, which seem obvious.
Comments on the Quality of English Language
Minor editing of English language required
Author Response
Dear Reviewer,
please find our reply to your report bellow, and as a word document in attachment too.
Thank you.
Best,
Tarik Dizdarević.
Reviewer 1:
The authors have done a good piece of work. However, a major revision is needed before publication. The paper lacks a detailed description of the cheese production technology, considering that this is an important aspect that can influence the microflora of the product, the authors should include a description of the production technology in the text.
Answer:
Dear reviewer,
we have done all necessary changes you proposed. As for cheese technology we didn't want to occupy to much space since one should be concise but we fully agree and understand that cheese production technology can influence the microflora of product. Therefore, we put into work description of cheese technology.
Lines 101-102: What are the differences between the 3 productions? Which seasons were taken into account?
Following cheese production processs and taking samples from three different producers and two seasons we tried to take into account differences due to -from day to day- variations in chemical and microbiological properties of cheese milk as well as cheese technology which normaly occour between producers and seasons. Two consecutive seasons were followed. The season of Livno cheese production is late spring-summer period.

Reviewer 2 Report
Comments and Suggestions for Authors
I appreciate this work being done over two seasons. However this type of research is not new; there are now hundreds of similar papers that have been published over the last 10 years or so.
I have serious concerns how mesophilic cocci (lactococci, pediococci) and thermophilic cocci (streptococci) were enumerated. Firstly M17 is not a selective agar medium for lactic acid bacteria, many bacteria can grow on it. There are also issues but less serious ones in using MRS even if it is acidified. My concern is that the authors show no awareness of these well known issues and have not controlled for them e.g. presumptive identification of colonies using basic microbiological tests.
It is disappointing that the authors have not adequately compared their results with the results of other similar studies.
Comments on the Quality of English Language
While the English could be improved it is satisfactory.
Author Response
Dear Reviewer,
please find our reply to your report bellow, and as a word document in attachment too.
Thank you.
Best,
Tarik Dizdarević.
Reviewer 2 – answers
Reviewer 2:
I appreciate this work being done over two seasons. However this type of research is not new; there are now hundreds of similar papers that have been published over the last 10 years or so.
I have serious concerns how mesophilic cocci (lactococci, pediococci) and thermophilic cocci (streptococci) were enumerated. Firstly M17 is not a selective agar medium for lactic acid bacteria, many bacteria can grow on it. There are also issues but less serious ones in using MRS even if it is acidified. My concern is that the authors show no awareness of these well known issues and have not controlled for them e.g. presumptive identification of colonies using basic microbiological tests.
It is disappointing that the authors have not adequately compared their results with the results of other similar studies.
Answer:
Dear reviewer,
we accept all suggestions and we have made all required changes and additions. We did comparation with results of other authors.
|
Line reference |
Comments from the referees |
Write the response |
|
|
I have serious concerns how mesophilic cocci (lactococci, pediococci) and thermophilic cocci (streptococci) were enumerated. Firstly M17 is not a selective agar medium for lactic acid bacteria, many bacteria can grow on it. There are also issues but less serious ones in using MRS even if it is acidified. My concern is that the authors show no awareness of these well known issues and have not controlled for them e.g. presumptive identification of colonies using basic microbiological tests.
|
We agree regarding selectivity of M17 and MRS agar, but in moment of realization of research that was the only media affordable to us. Also this is an area of slight uncertainty. We tried to respect your comment and suggestion so we corrected mesophilic cocci (including lactococci, pediococci) and thermophilic cocci (streptococci) to presumptive presumptive mesophilic and thermophilic streptococci and presumptive mesophilic and thermophilic lactobacilli |
|
|
It is disappointing that the authors have not adequately compared their results with the results of other similar studies. |
We added available results of similar studies. |

Reviewer 3 Report
Comments and Suggestions for Authors
The objective of this research was to examine the microbial biodiversity of the traditional Livno raw milk cheese made by various producers, during ripening and from two production seasons using culture-dependent and independent methods.
It is interesting to know the microbial composition of this traditional cheese, which does not use microbial starters, to identify the dominant microorganisms and develop an autochthonous starter. Highlighting the use of two different molecular methods (DGGE and ARISA) to determine the microbial compositions and compare it to culture methods is interesting.
Nevertheless, I would have appreciated it if the authors had included the determination of physicochemical or sensorial properties of the cheese of the different producers to link some typical characteristics of these cheeses with the abundance of different bacteria groups.
Below you will find some comments to improve the manuscript:
Abstract
L24. Clarify which culture-independent methods are used in the present research
L26. At the end, a summary of the advantages and disadvantages of the use of culture-dependent and culture-independent methods in the evaluation of Livno cheese microbiota… as the not detection of Enterococcus by culture-independent methods.
Introduction
L62-63. Correct microorganisms' names by adding cursive and upper letter
Material and methods
L112. How was it plated?
L119. Explain better the type of samples…Livno cheese curd, fresh salted cheese, and cheeses during ripening.
L121. Remove point.
L124. Add incubation temperature and time of aerobic bacteria enumeration.
L128. Correct the degree symbol, and review it in all the document
L129. Add point
L130. Correct Enterococci to Enterococcus
L131. Add the number of samples that were analysed at each sampling time. n?
L150. Replace spp. by subsp.
L155. Remove point
Results
L201. Replace by culture-dependent methods.
L207. What is the meaning of the asterisk?
Table 1. What is the distribution of microbial enumeration in each sampling time?
Correct Enterococci to Enterococcus
In data, replace comma by point
L235. Correct number of table
Table 2. Correct identification
L239. Review microorganism name
L245. Correct format
L250. Revise writing
L273. Correct PCR_DGGE
Figure 2. Revise the format of the index, include a space between genera and species, and correct the name of Enterococcus
L287. Revise the sentence because the 1st season presents a higher presence of Lc. garvieae than in the 2nd season.
L294. Specify if you are talking there about the last days of ripening because in the curd a higher presence of S. thermophilus we observe in producer B.
L296. Remove point
L299. Could you refer to L. plantarum in this sentence?
L304. Add cursive
L303-306. Detail that it occurred in the 1st season
L308. Highlight the increase of S. thermophilus during ripening in the 2nd season
L309-310. Could you refer to S. faecium in this sentence?
Discussion
L319. Does it mean at the end of 90 days of ripening?
L320. Explain if the value is the average of the 3 producers
L324. It is not the correct abbreviation
L331. Remove point
L342. It is necessary to describe the initials the first time that appear in the text
L373. Enterococci to Enterococcus
L387. Correct logarithmic units
L393. Add cursive
L395. Correct E. fecium to E. faecium
L423. Correct lover
L435. Replace spp. by subsp.
L445. Develop more the motive why was Enterococcus not detected.
L484. Add cursive
L494. Add a point
L504. I think that you refer there to cultivation-dependent methods.
L515. Remove point
L518. Remove one point

Author Response
Dear Reviewer,
please find our reply to your report bellow, and as a word document in attachment too.
Thank you.
Best,
Tarik Dizdarević.
Reviewer 3:
The objective of this research was to examine the microbial biodiversity of the traditional Livno raw milk cheese made by various producers, during ripening and from two production seasons using culture-dependent and independent methods.
It is interesting to know the microbial composition of this traditional cheese, which does not use microbial starters, to identify the dominant microorganisms and develop an autochthonous starter. Highlighting the use of two different molecular methods (DGGE and ARISA) to determine the microbial compositions and compare it to culture methods is interesting.
Nevertheless, I would have appreciated it if the authors had included the determination of physicochemical or sensorial properties of the cheese of the different producers to link some typical characteristics of these cheeses with the abundance of different bacteria groups.
Answer:
We agreed with the last part but we didn't include it in this manuscript because of limited space and high quantity of results. Here we focused on studying of microbial biodiversity. A comprehensive research as we did requires more than one paper to elaborate all results and we feel that more results of different type to the ones we have presented would jeopardize the clarity of investigation results. We find reviewers thinking reasonable and logical and hope this explanation is justified, clear and understandable.
|
Line reference |
Comments from the referees |
Write the response (Corrected/ Changed/ Addition made as suggested/ Rephrased to make more clear) |
|
Abstract |
|
|
|
L24. |
Clarify which culture-independent methods are used in the present research |
Rephrased to make more clear |
|
L26. |
At the end, a summary of the advantages and disadvantages of the use of culture-dependent and culture-independent methods in the evaluation of Livno cheese microbiota… as the not detection of Enterococcus by culture-independent methods. |
Done |
|
Introduction |
|
|
|
L62-63. |
Correct microorganisms' names by adding cursive and upper letter |
Corrected |
|
Material and methods |
|
|
|
L112. |
How was it plated? |
Corrected |
|
L119. |
Explain better the type of samples…Livno cheese curd, fresh salted cheese, and cheeses during ripening. |
Corrected/ Rephrased to make more clear L129. A total of 30 samples of traditional Livno cheese and curd were taken from three different producers (A, B, C) …… |
|
L121. |
Remove point. |
Corrected/ |
|
L124. |
Add incubation temperature and time of aerobic bacteria enumeration. |
Addition made as suggested “for 72 h at 30 °C” |
|
L128. |
Correct the degree symbol, and review it in all the document |
Correct in all the document |
|
L129. |
Add point |
Corrected |
|
L130. |
Correct Enterococci to Enterococcus |
Corrected |
|
L131. |
Add the number of samples that were analysed at each sampling time. n? |
Corrected. …for each sampling time 6 samples were analyzed. |
|
L150. |
Replace spp. by subsp. |
Corrected |
|
L155. |
Remove point |
Corrected |
|
Results |
|
|
|
L201. |
Replace by culture-dependent methods. |
Changed |
|
L207. |
What is the meaning of the asterisk? |
Corrected |
|
Table 1. |
What is the distribution of microbial enumeration in each sampling time?
Correct Enterococci to Enterococcus
In data, replace comma by point |
Corrected
Unfortunately, we were unable to change the comma to a full stop. The Excel spreadsheet that was used to calculate log 10 does not allow us to change it |
|
L235. |
Correct number of table |
Corrected |
|
Table 2. |
Correct identification |
Corrected |
|
L239. |
Review microorganism name |
Corrected |
|
L245. |
Correct format |
Corrected |
|
L250. |
Revise writing |
Revised |
|
L273. |
Correct PCR_DGGE |
Corrected |
|
Figure 2. |
Revise the format of the index, include a space between genera and species, and correct the name of Enterococcus |
Corrected |
|
L287. |
Revise the sentence because the 1st season presents a higher presence of Lc. garvieae than in the 2nd season. |
Changed ….dominance of Lc. garvieae compared to the 2nd season |
|
L294. |
Specify if you are talking there about the last days of ripening because in the curd a higher presence of S. thermophilus we observe in producer B. |
Corrected
|
|
L296. |
Remove point |
Corrected
|
|
L299. |
Could you refer to L. plantarum in this sentence? |
Finished |
|
L304. |
Add cursive |
Corrected |
|
L303-306. |
Detail that it occurred in the 1st season |
Finished |
|
L308. |
Highlight the increase of S. thermophilus during ripening in the 2nd season |
Corrected |
|
L309-310. |
Could you refer to S. faecium in this sentence? |
We didn’t find this |
|
Discussion |
|
|
|
L319. |
Does it mean at the end of 90 days of ripening? |
Changed ….increase from day 1 to the end of ripening at 90 days |
|
L320. |
Explain if the value is the average of the 3 producers |
Corrected…..The averages of microorganisms enumerated (log10 CFU·g-1) in traditional Livno cheese of the 3 producers for |
|
L324. |
It is not the correct abbreviation |
Changed ….SB agar |
|
L331. |
Remove point |
Corrected |
|
L342. |
It is necessary to describe the initials the first time that appear in the text |
Corrected |
|
L373. |
Enterococci to Enterococcus |
Corrected |
|
L387. |
Correct logarithmic units |
Corrected |
|
L393. |
Add cursive |
Corrected |
|
L395. |
Correct E. fecium to E. faecium |
Corrected |
|
L423. |
Correct lover |
Corrected |
|
445. |
Develop more the motive why was Enterococcus not detected. |
Developed |
|
L484. |
Add cursive |
Corrected |
|
L494. |
Add a point |
Corrected |
|
L504. |
I think that you refer there to cultivation-dependent methods. |
Corrected |
|
L515. |
Remove point |
Corrected |
|
L518. |
Remove one point |
Corrected |

Round 2
Reviewer 1 Report
Comments and Suggestions for Authors
The authors revised the article and corrected all my comments. I believe that the article can be accepted for publication in its current form.
Comments on the Quality of English Language
Moderate editing of English language required
Reviewer 2 Report
Comments and Suggestions for Authors
I have serious concerns about the cultural studies. Other aspects are good. Unless they can conform that they have done the necessary work to confirm their isolates there is a real problem with this paper.
Work is not novel but this is an Editorial issue.
Author Response
Dear Reviewer 2.
Different nutrient media and incubation temperatures were used for the isolation of 1200 isolates of bacteria from traditional Livno cheese.
During our experiment, we did the Catalase test and Gram staining. After number of 1200 isolates was reduced on 360. All 360 isolates were tested on:
- ability to produce CO2,
- the growth ability at different temperatures (10 ºC and 45 ºC)
- the growth ability at different salt concentrations (2%, 4.5% and 6.5% NaCl).
Also technological characterization of the isolated strains (results not shown) was examined based on the acidification rate of the isolates in milk. From a total of 360 isolates was seelected 200 microbial isolates and sequenced with 16 S rRNA.
From the total number of sequenced isolates, it was determined that the dominant microbial groups of Livno cheese were enterococci (36.50%), lactobacilli (31.00%), pediococci (11.50%), lactococci (8.50%), streptococci ( 7.00%) and leuconostoc (5.50%).
About vancomycin concentration use for Leuconostoc spp. we used MRS with addition of 20 µg/ml (20 mg/l) vancomycin.
Best,
Reviewer 3 Report
Comments and Suggestions for Authors
After introducing my suggestions in the manuscript and receiving your allegations, I have considered to accept your manuscript.
Only, I have identified some text editing mistakes:
L189: “itial denaturing program 95 °C for 5 min, 30 denaturation cycles at 95 °C for 30 sec, an-”. add the correct sec abbreviation
L438: “by PCR: E. faecium (71.2%), E. durans (20, 3%) and E. faecalis (8.5%) On the contrary, re-”. Change comma to a full stop in the value and add full stop at the end of the sentence.
Author Response
Dear Reviewer 3., text editing mistakes were corrected